# Sporadic low-velocity volumes spatially correlate with shallow very low frequency earthquake clusters

Takashi Tonegawa[1], Eiichiro Araki[1], Toshinori Kimura[1], Takeshi Nakamura[2], Masaru Nakano [1] & Kensuke Suzuki[1]

A low-velocity zone (LVZ) has been detected by seismic exploration surveys within the Nankai accretionary prism toe off the Kii Peninsula, southwestern Japan, and is considered to be a mechanically weak volume at depth. Such mechanical heterogeneities potentially influence seismic and tsunamigenic slips on megathrust earthquakes in the subduction zone. However, the spatial distribution of the LVZ along the trough-parallel direction is still elusive. Here we show sporadic LVZs in the prism toe from one-dimensional shear wave velocity (Vs) profiles obtained at 49 cabled ocean bottom stations, which were estimated by a nonlinear inversion technique, simulated annealing, using the displacement–pressure ratios of the Rayleigh wave. The estimated distribution of LVZs along the trough widely correlates with the epicentres of shallow very low frequency earthquakes (sVLFEs), which suggests that sVLFEs are activated in the sporadically distributed low-velocity and mechanically weak volumes where fluids significantly reduce the shear strength of faults.

[1] Japan Agency for Marine-Earth Science and Technology, 2-15, Natsushima, Yokosuka, Kanagawa 237-0061, Japan. [2] National Research Institute for Earth Science and Disaster Resilience, 3-1, Tennodai, Tsukuba, Ibaraki 305-0006, Japan. Correspondence and requests for materials should be addressed to T.T. (email: tonegawa@jamstec.go.jp)

At accretionary prism toes in subduction zones, fluid-rich conditions are induced by water released from the marine sediments and oceanic crust of the subducting oceanic plates. The fluid distribution within prism toes depends on several factors, including fluid source locations, fluid generation rate, and spatial permeability variations[1]. The concentration of fluid leads to a condition of elevated pore pressure at localized regions. Such a condition potentially weakens the shear strength of faults by reducing the effective stress and hence has been linked to slow earthquakes occurring within the prism toe, including shallow very low frequency earthquakes (sVLFEs)[2,3], which show a lower stress drop than ordinary earthquakes[4], and low-frequency tremors[5,6].

In fluid-concentrated areas, seismic velocity is significantly reduced relative to that in surrounding dry areas. A thin, low P-wave velocity (Vp) zone (LVZ) near plate convergent margins has been detected by seismic surveys in other parts of the world, such as in an accretionary wedge in Costa Rica (< 400 m thick)[7] and around the plate interface at shallow depths in Ecuador (~ 600 m thick)[8], and fluid concentration is considered a candidate for the cause of the observed LVZs. Within the toe of the Nankai accretionary prism in southwestern Japan, an LVZ with a maximum thickness of ~ 2 km was found using two sophisticated seismic approaches, including a three-dimensional prestack depth migration[9] and a full waveform inversion[10]. The Vp varies from 1.6 to 3.5 km s$^{-1}$, to 2.7 to 3.2 km s$^{-1}$ within the LVZ and overlying layers because of antiformal stacking of underthrust sediments overriding the décollement[9]. As pore fluid pressure supports more than half of the overburden stress, the LVZ in the Nankai region is thought to be a mechanically weak volume at depth[11]. Therefore, LVZ exploration in the prism toe along the margin-parallel direction in the Nankai subduction zone is key to understanding the distribution of the mechanically weak volumes that potentially influence coseismic rupture propagation and tsunamigenic slip on megathrust earthquakes. Indeed, devastating large earthquakes have historically occurred along this subduction zone, most recently the 1944 Tonankai ($M_w$ = 8.1) and 1946 Nankai ($M_w$ = 8.4) earthquakes[12] (Fig. 1). Nevertheless, as the locations of the two seismic surveys are close to each other[9,10] and

they only resolve the LVZ structure along the margin-normal direction, whether the LVZ is localized there or distributed along the margin-parallel direction is still unknown.

Off the Kii Peninsula, a cabled network, the Dense Oceanfloor Network System for Earthquakes and Tsunamis (DONET), has been deployed for monitoring seismic activity (Fig. 1 and Supplementary Fig. 1)[13,14]. The network consists of 51 stations, of which 22 DONET1 stations and 29 DONET2 stations are deployed southeast and southwest of the Kii Peninsula, respectively, providing extensive coverage of the accretionary prism from the trough, i.e., 300 km in the trough-parallel direction and 100 km in the trough-normal direction, with a station spacing of 15–20 km. The observation periods of DONET1 and 2 exceed more than 5 years and 1 year, respectively (as of 1 April 2017).

In this study, we attempt to estimate the one-dimensional shear wave velocity (Vs) structure beneath each station using the Rayleigh admittance (RA)[15] calculated from records observed at the DONET stations. The RA is an amplitude transfer function between the displacement and pressure fluctuation (subtracting the mean) observed at the seafloor during Rayleigh wave propagation, and is sensitive to shallow Vs structure beneath the seafloor (< 10 km depth, see Methods). Therefore, the RA is capable of exploring the LVZ that was detected at shallow depths in the prism toe[9,10]. We apply a nonlinear inversion technique, i.e., simulated annealing[16], to observed RAs for estimating Vs profiles at each station (Methods). To ensure the reliability of the obtained results, we use two types of RAs: (1) the RA calculated from earthquake-excited Rayleigh waves (e-RA) and (2) the RA connecting the e-RA in the lower frequency components with the RA from ambient noise records at the frequency band of microseisms (n-RA) in the higher frequency components (en-RA, see Methods); we then confirm the consistency of the two results at a single station. Using obtained Vs profiles, we explore the LVZ along the trough-parallel direction in the Nankai accretionary prism toe.

## Results

**Vs structure and LVZ distribution.** Figure 2 displays Vs profiles beneath each station along L1–L4 (Fig. 1). A simple Vs structure can be seen at stations near land, i.e., Vs increases with increasing depth, whereas some of the Vs profiles at the prism toe exhibit an LVZ at depths of 4–9 km below sea level (kmbsl) (Fig. 2a, b, d). Here, we defined the LVZ feature as a Vs profile with a velocity reduction > 20% in the depth range of 5–10 kmbsl from a reference velocity model averaged over the obtained Vs profiles beneath the seafloor, irrespective of Vs gradient as a function of depth (including a lower Vs layer compared with those above and below the layer, and a constant or increasing Vs layer). This means that the LVZ feature indicates a significant low Vs layer in the prism toe compared with the Vs averaged over the entire accretionary prism, and we plot the Vs profiles including the LVZ feature with colored lines (Fig. 2 and Supplementary Fig. 2). If such a velocity structure is identified at stations away from the prism toe landward and directly below the seafloor, it is not related to the LVZ in the prism toe. The features of the Vs profiles with and without the LVZ feature estimated from en-RAs and e-RAs are almost consistent (Fig. 2 and Supplementary Fig. 2), and the SD of the obtained LVZ features ensures the reliability of the velocity reduction (Supplementary Fig. 2). The location of the LVZ detected by previous studies[9,10] coincides with node KMD (See the node and station names in Fig. 1 and Supplementary Fig. 1) southeast of the Kii Peninsula, and our Vs profiles at the southern three stations at node KMD also exhibit the LVZ feature (Fig. 2d). As two different décollement locations have been

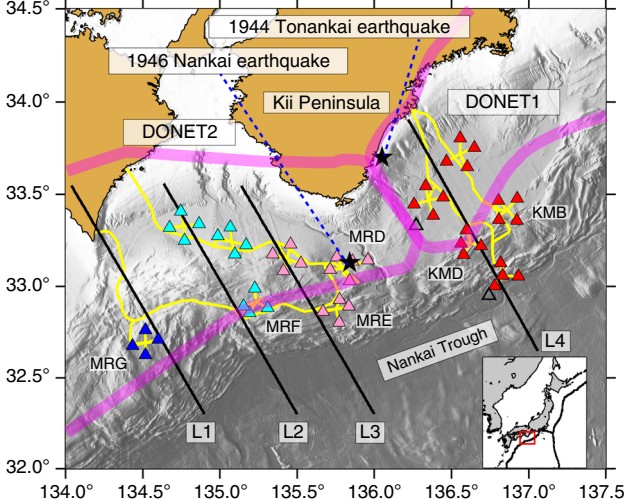

**Fig. 1** Locations of stations in DONET1 and DONET2, and lines. Red triangles are DONET1 stations, and blue, pale blue, and pink triangles are DONET2 stations. Labels with three capital letters represent the node name (see details in Supplementary Fig. 1). Vs profiles at the stations are projected onto lines L1–L4 in Fig. 2. Stars represent the locations of rupture initiations for the 1944 Tonankai and 1946 Nankai earthquakes[12], and magenta line shows seismogenic zones for future great earthquakes

suggested in this area[17,18], we avoid mentioning the LVZ location with respect to the décollement and conclude that the LVZ is located within the accretionary prism toe. In addition to this area, our Vs profiles at stations southwest of the Kii Peninsula also display the LVZ feature (Fig. 2a, b). Based on the depth from the seafloor and distance from the trough, the LVZ should be present within the accretionary prism beneath nodes MRF and MRG, as is the case beneath node KMD.

Figure 3 shows the minimum Vs perturbations at 5–10 kmbsl at each station from a reference Vs profile averaged over the Vs profiles beneath the seafloor for all of the employed stations, which indicates that large velocity reductions (>20%) are present

in the prism toe (at nodes KMB, KMD, MRF, and MRG), mainly at stations close to the trough inside the nodes. In particular, a patch-like low-velocity region can be seen southeast of the Kii Peninsula. On the other hand, although the station locations of nodes MRD and MRE cover the entire prism toe, no significant LVZ feature was observed (Figs. 2c and 3, and Supplementary Fig. 3). Our numerical tests indicate that the RA is insensitive to the velocity change in a thin layer (< 1 km) and to the velocity change in a deep structure (> 10 km) (Methods). Therefore, we consider that the absence of the LVZ feature south of the Kii Peninsula indicates that no LVZ or only a very thin LVZ is present in the prism toe, and that the observed LVZ feature

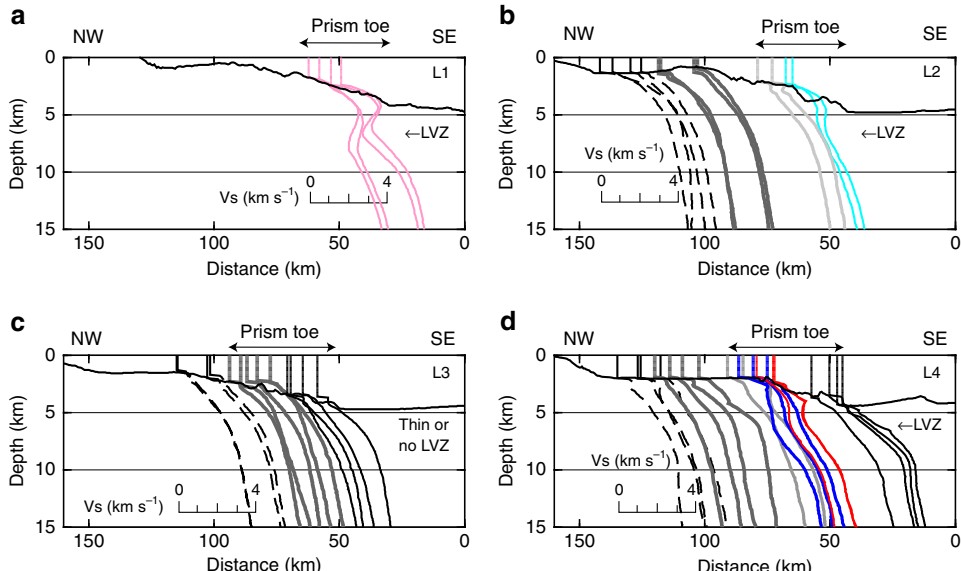

**Fig. 2** Vs profiles for each line. Vs profiles for **a** L1, **b** L2, **c** L3, and **d** L4 (Fig. 1). Station locations (Fig. 1) are projected onto L1–L4 and corresponding Vs profiles are plotted at the projected location. Pink, pale blue, blue, and red lines represent the LVZ feature at nodes MRG, MRF, KMB, and KMD, respectively. Black solid, dashed, and gray lines correspond to Vs profiles at stations belonging to the same nodes, but without the LVZ feature. Light gray lines in **d** correspond to Vs profiles at stations KMD16 (right) and KMB08 (left) (Supplementary Fig. 1), without the LVZ feature. Solid line at each panel indicates bathymetry along each line

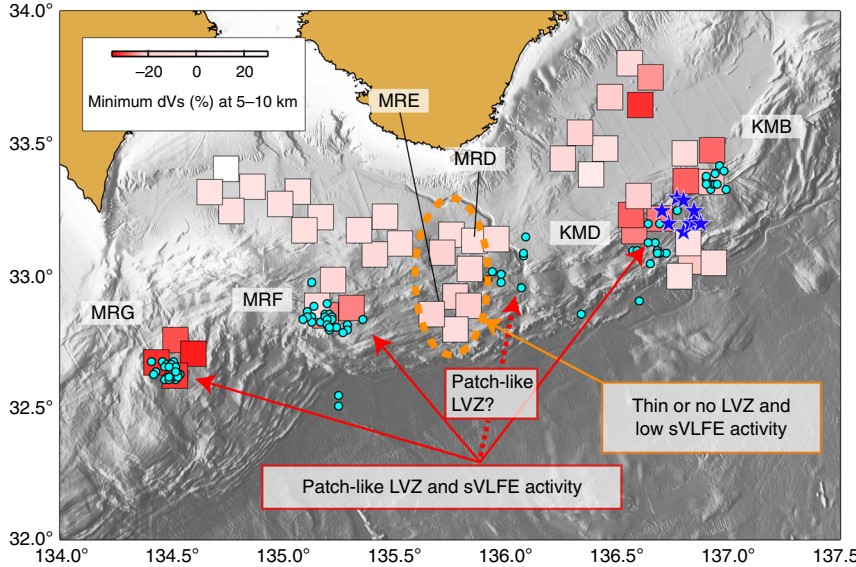

**Fig. 3** Spatial relationship between LVZ and sVLFE determined by seafloor records. The intensity of red in the squares indicates the minimum dVs in the depth range of 5–10 kmbsl. Locations of squares correspond to station locations. Pale blue circles[29] and blue stars[23] show epicentres of sVLFEs determined from seafloor records

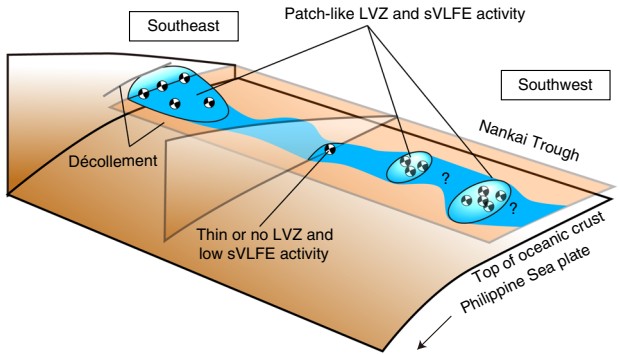

**Fig. 4** Schematic for scattered LVZs in Nankai accretionary prism toe. The LVZs are sparsely distributed in the prism toe and the shear strength of faults inside the LVZs is weakened by the presence of fluid. The occurrence of sVLFEs may be promoted in the LVZ. Gray line[18] and plane[17] indicate two interpretation of the décollement location according to previous studies

southeast and southwest of the Kii Peninsula indicates the presence of a thick LVZ (> 1 km).

**LVZ as an elevated pore pressure region**. The estimation of the pore pressure ratio from the seismic velocity structure would be informative for evaluating the mechanical strength of the LVZ. The pore pressure ratio ($\lambda^*$) represents the degree of pore fluid pressure that supports the overburden stress due to the lithostatic load, where $\lambda^* = [(\text{pore pressure} - \text{hydrostatic pressure})/(\text{lithostatic pressure} - \text{hydrostatic pressure})]$ ($\lambda^* = 0$ indicates no pressure contributions of pore fluid, and $\lambda^* = 1$ indicates lithostatic pore pressure)[1,19]. For the LVZ southeast of the Kii Peninsula, $\lambda^*$ was estimated to be 0.54–0.77 from a Vp structure in which fluids support 74–87% of the overburden stress[11,20]. Another previous study[21] estimated $\lambda^*$ within the prism toe in the same location from Vp and Vs structures, and argued that, although the $\lambda^*$ from the Vs structure is slightly higher than that from the Vp structure, the $\lambda^*$ distributions are similar to each other, and they are also consistent with the result from the previous study[11].

Using the method of a previous study[11] with an empirical relationship between Vp and Vs[22], we also estimated profiles of pore pressure ratios beneath the DONET stations for a calibration of $\lambda^*$ with respect to that of the previous study at the southeast of the Kii Peninsula and an investigation of the lateral variation of $\lambda^*$ within the LVZs in the Nankai accretionary prism toe. Supplementary Fig. 4 shows one-dimensional profiles of pore pressure ratio at each station. For profiles at the stations where an LVZ was found, high pore pressure ratios ($\lambda^* = 0.70$, 0.69, and 0.75 at nodes KMD, MRF, and MRG, respectively, at 6.5 kmbsl: the LVZ feature emerges at this depth at most stations) can be seen, such that fluids support 82–85% of the overburden stress. The pore pressure ratio in the southeast of the Kii Peninsula is in good agreement with previous estimations[11,20], and similar and slightly higher $\lambda^*$ values were obtained in the LVZs in the southwest of the Kii Peninsula. As in a previous study[11], the hypocentres of sVLFEs[23] showing low stress drops[4] coincide with the high pore pressure region along L4. On the other hand, the pore pressure ratios for profiles at nodes MRD and MRE appear relatively low ($\lambda^* = 0.35$–0.49 at 6.5 kmbsl, 65–73% of the overburden), except for two profiles (MRD17 and MRE18: $\lambda^* = 0.54$ and 0.59 at 6.5 kmbsl, 75 and 78% of the overburden) at horizontal distances of 50 and 30 km from the trough, respectively.

The sediment and oceanic crust of the incoming plate carry pore water in cracks and bound water in hydrous minerals into subduction zones, and most of the pore water is expelled by

horizontal compaction near the plate convergent margin[1,19,24]. Dehydration reactions in the sediment and oceanic crust tend to dominate as major fluid sources behind the region where the effect of horizontal compaction is dominant for releasing fluid[1,24]. The LVZ-related fluid would be presumably provided by these two mechanisms.

The underlying formation mechanisms of the heterogeneous LVZ remain elusive. The LVZ may have been formed by fluid being trapped by overriding undeformed sediments that act as a structural seal for upward fluid migration[10]. This implies that the highly deformed sediments in the prism toe may allow fluids to migrate upward without being trapped. Alternatively, if shear fractures are created by deformation associated with high sVLFE activity, they potentially trap fluids, hence producing the LVZ. For further investigation of fluid-trapping mechanisms in the LVZ, integration of seismic exploration surveys focusing on the degree of sediment deformation in the Nankai accretionary prism toe would be required.

**Spatial relationship between LVZ and sVLFE**. The sVLFE activity along the trough has been observed to be spatially distributed in a sporadic manner, with some clusters[3,25]. According to long-term, land-based observations, the activity at each cluster is repeated temporally with an interval of a few years[26,27]. This activity is triggered after large earthquakes[3,26–28], but spontaneous activity without such triggering is also observed[2,3,23,29].

Figure 3 shows a comparison of the distribution of the LVZ and sVLFE activity around the Nankai Trough from two catalogs[23,29] determined by seafloor-based observations around the Nankai Trough. This figure shows some clusters of sVLFEs in the prism toe. As these catalogues provide sVLFE epicentres with high spatial resolution, and one catalog[23] determines the focal depths of sVLFEs of 5.2–11.6 km with a realistic velocity model, it appears that VLFEs occur at shallow depths in the prism toe.

As shown in Fig. 3, the regions of sVLFE activity coincide with the locations of the thick LVZ southeast and southwest of the Kii Peninsula. Our Vs profiles cannot identify the spatial extent of the LVZ because of lack of station coverage. However, we suppose that the size of the LVZs are localized within the prism toe from the following observations: (1) as shown in Fig. 3, LVZs can be seen at all stations at node MRG, whereas some of the stations at nodes KMB, KMD, and MRF shows such features, and (2) the degree of low Vp along the trough-parallel and trough-normal directions varies within the LVZ[9]. Moreover, based on both the station locations with low Vs (red squares in Fig. 3) and the spatial extent of each sVLFE cluster determined from seafloor records, we consider that patch-like LVZs are distributed in the Nankai accretionary prism (Figs. 3 and 4) and the spatial correlation between the LVZ and sVLFE indicates that the high sVLFE activity is activated in the patch-like mechanically weak volumes where fluid concentration reduces the shear strength of faults.

Supplementary Fig. 3 shows another catalog[27] determined by land-based observations, which contains a long-term history of VLFE activity in 2003–2016. This catalog offers highly accurate epicentres in the trough-parallel direction because of land-based determinations with sufficient azimuthal coverage data. A scattered distribution of the sVLFE epicentres along the trough-normal direction is caused by a low spatial resolution along that direction. Based on the concentration of sVLFEs in the prism toe in the catalogs from the seafloor-based observations[23,29], it appears that these events determined from land-based observations occur at shallow depths in the prism toe. As shown in Supplementary Fig. 3, although sVLFEs have occurred south of

the Kii Peninsula[3,29], the activity in the long-term catalog appears low with respect to that on either side of the region[26,27]. As the locations of the sVLFE[29] are somewhat distinct from the two nodes, MRD and MRE, a patch-like LVZ may be present southeast of the two nodes (Fig. 4). Otherwise, a localized region with relatively high $\lambda^\star$ is present, as seen at nodes MRD and MRE, and it may contribute to the occurrences of a small number of sVLFEs.

## Discussion

Previous studies found the presence of an LVZ in the accretionary prism toe southeast of the Kii Peninsula[9,10,22] and it is characterized as a mechanically weak volume at depth[11]. Land-based observations have elucidated that the epicentres of sVLFEs are sporadically distributed along the Nankai Trough[3,25]. However, the spatial relationship between the sVLFE and LVZ is unknown. In this study, we present one-dimensional Vs profiles beneath 49 cabled ocean bottom stations in DONET by applying a nonlinear inversion technique, simulated annealing, to the RA. Our result shows that patch-like LVZs in the prism toe along the Nankai Trough are correlated with the locations of the sVLFE clusters and also that VLFE activity is low in a region where the LVZ feature cannot be seen. We therefore interpret that sVLFEs may be activated by the reduction in the faults' shear strength because of fluid within the spotted LVZs. Moreover, this distribution of LVZs, characterized as a mechanically weak volume at depth, could affect the rupture propagation and tsunamigenesis of future megathrust earthquakes.

Evidence of sVLFE activities has been found at subduction zones around Japan[25–27,30–33] and Costa Rica[34]. The lack of sVLFE observations at other subduction zones would be caused by either (1) no sVLFEs occurring at other subduction zones or (2) sVLFE activity occurring intermittently in time and sporadically in space. In the latter case, long-term seafloor observations with dense array are required to capture sVLFE signals and also thick LVZ exploration in accretionary subduction zones all over the world may be useful for searching future sVLFE activities.

## Methods

**Shear wave velocity structure from RA.** In order to estimate the subseafloor seismic velocity structure, several effective approaches have been implemented using seafloor records, such as surface wave analysis using ambient noise correlation and/or earthquake-excited signals[35–38] and receiver functions[39–41]. In this study, we chose an RA analysis[15] to explore the LVZ for the following two reasons: (1) the degree of velocity reduction in the LVZ may vary spatially and single-station analysis is suitable for estimating velocity structure; (2) determination of the absolute Vs value is required for quantitatively evaluating the Vs value with consideration of fluid effects.

**Seismograms and pressure records.** We used displacement and pressure records observed at DONET1 and DONET2, which consist of five and seven nodes, respectively, and each node contains four or five stations. The water depths at which the stations are deployed range ~ 1,900–4,400 m in DONET1 and 1,000–3,600 m in DONET2. Each station has a broadband seismometer (Güralp CMG-3T, flat velocity response from 50 Hz to 360 s)[42], an absolute pressure gauge (APG) (Paroscientifix Inc. 8B7000-2)[43], and a differential pressure gauge (DPG). The sampling rate of these records is 200 Hz.

Compared with the APG, the DPG is capable of observing the pressure fluctuation with higher resolution, but its instrumental response is not known. Therefore, using earthquake signals, we estimated the transfer function, including the amplitude and phase between the APG and DPG at each station. The transfer function calculation is summarized elsewhere[44]. We chose earthquakes with epicentral distances of 15–90° and magnitudes > 7 for DONET1 and 6.5 for DONET2. The reason for the smaller magnitude criterion for DONET2 is to collect a sufficient number of Rayleigh wave records in the relatively shorter observation period of DONET2. The mean amplitude in both the APG and DPG records are subtracted in the time domain. Using a time window of − 50 to + 250 s with respect to the P arrival for the APG and DPG records, the relative amplitude and phase are calculated in the frequency domain (Supplementary Fig. 5). In order to use the relative information for correcting DPG records, we fit them with a quadratic function and estimate the coefficients. The number of earthquakes used in this

analysis is summarized in Supplementary Table 1. In DONET1, stations KMC21 and KME22 were newly deployed. For these two stations, because the number of available earthquakes was insufficient, we could not perform the Vs structure estimation (Fig. 1).

**Preparation of RA.** The RA $\eta(f)$ can be written as

$$\eta(f) = \left| \frac{u_z(f)}{\Delta P(f)} \right|, \tag{1}$$

where $u_z(f)$ is the vertical displacement, $\Delta P(f)$ is the pressure fluctuation with the mean subtracted, and $f$ is the frequency[15]. The displacement and pressure fluctuation are observed at the seafloor when the Rayleigh wave propagates to the station[15]. In this study, we prepared RAs from the fundamental Rayleigh mode associated with earthquakes and microseisms, and connected them at the frequency at which the amplitude of the Rayleigh wave in the microseisms becomes weak, following the method of a previous study[15]. The displacement records were calculated by removing the instrumental response from the vertical velocity seismogram observed at the broadband seismometers in DONET. The amplitude and phase of the DPG records were corrected using the quadratic functions estimated in the previous section. We call the RA from Rayleigh waves associated with earthquakes and microseisms in ambient noise e-RA and n-RA, respectively, and also call the RA connecting the e-RA at lower frequency with n-RA at higher frequency en-RA.

For e-RAs, we collected seismograms and DPG records of Rayleigh waves from earthquakes with magnitudes > 5.0, epicentral distances of 15–90°, and focal depths shallower than 50 km. The total number of earthquakes at each station is summarized in Supplementary Table 1. We examined the time duration of 800 s from the Rayleigh wave arrival. For event selection, we calculate the coherence averaged over frequencies of 0.03–0.09 Hz between the displacement and the DPG spectra, and discard e-RAs < 0.9. The obtained e-RAs are smoothed by the Parzen window with a frequency band of ± 0.01 Hz.

For the estimation of n-RAs, the transfer function was calculated with continuous records of displacement and the DPG with a time duration of 600 s. In order to remove energetic signals, including earthquakes, we calculated four quantities using time series in the vertical displacement component: (1) the root-mean square (RMS) amplitude with time durations of 600 s at a frequency of 2–4 Hz ($A_1$); (2) the RMS with a time window of 3,600 s, surrounding the time window of 600 s in $A_1$, at a frequency of 2–4 Hz ($A_2$); (3) the same as $A_1$, but for 0.03–0.06 Hz ($B_1$); and (4) the same as $A_2$, but for 0.03–0.06 Hz ($B_2$). These two frequency bands correspond to relatively low levels of ambient noise. If $A_1/A_2 > 3$ or $B_1/B_2 > 3$, we discard the record, because the time series may contain energetic signals. The transfer functions are stacked over 1 day and smoothed by the Parzen window with a frequency band of ±0.01 Hz.

To connect the e-RA and n-RA, it is necessary to determine the frequency band of the fundamental Rayleigh mode in the ambient noise observed at the DONET stations. As introduced by a previous study[15], we estimated the frequency band at which the coherence of the displacement and pressure records is > 0.9. Here we defined the lower- and higher-frequency limits as $fr_1$ and $fr_2$, respectively (Supplementary Fig. 5). In this study, we prepared an en-RA that composed of n-RA at frequencies between $fr_1$ and $fr_2$ and e-RA at frequencies lower than $fr_1$.

**Sensitivity of RA to physical parameters.** Before inverting the RA, we evaluated what parameters significantly vary the RA through calculations of synthetic RA. The displacement and stress eigenfunctions were calculated using DISPER80[45]. We used the Vs profile at station KMB06 (in 2,499 m water depth) of DONET1 as a reference velocity model: a Vp profile at each station can be constructed by referring to a P-wave tomographic velocity model[46], and Vs and density profiles are also created through empirical relationships from the Vp model[22]. To investigate the RA sensitivity, we tested the following six cases; (A) Vp reduction, (B) Vs reduction, (C) density reduction, (D) Vs reductions at various depths, (E) a water-depth change of − 2 m, and (F) acoustic velocity reduction near the sea surface. The assigned perturbations are summarized in Supplementary Table 2.

For models A–C, the three physical parameters, Vp, Vs, and density, are reduced within ± 1 km around a center depth of 4 km from the sea surface (Supplementary Fig. 6). The maximum reduction is 2% in the three parameters. As a result, the RA at frequencies higher than 0.1 Hz is changed by 2% for the Vs reductions and < 1% for the Vp and density reductions (Supplementary Fig. 6e) from the original RA (Supplementary Fig. 6d). A large variation in the RA for the Vp reductions can be seen at 0.11 Hz (blue line in Supplementary Fig. 6e) and this sensitive frequency is lower than those for Vs and density (red and black lines in Supplementary Fig. 6e).

In model D, we assigned a maximum reduction of 2% in Vs at 4, 6, and 9 kmbsl (Supplementary Fig. 6b). Supplementary Fig. 6f shows that, by increasing the depth at which the Vs reduction is maximum, the amount of RA variation decreases, and its sensitive frequency is reduced. This indicates that the RA tends to be insensitive to deep velocity structures. In model E, we examined a situation where the water-depth of a sensor changes during a long-term observation. The change might be caused by several events, such as large earthquakes and submarine landslides. In this case, we vertically uplifted a sensor by 2 m and replaced the physical

parameters of seawater with those of marine sediment. This resulted in a 0.2% in RA variation at most, but the variation extended over all frequencies (Supplementary Fig. 6g). Model F tested the effect of seasonal changes in acoustic velocity near the sea surface due to ocean currents. The maximum velocity reduction of 0.05 km s$^{-1}$ (3.3%) was assigned to the sea surface, and its reduction linearly decreased down to 0.2 km (Supplementary Fig. 6c). However, the effect on the RA variation was small (Supplementary Fig. 6g).

**Inversion of RA**. To estimate one-dimensional Vs profiles from the RA, we employed a nonlinear inversion technique, i.e., simulated annealing. The misfit function is described as

$$E = w_1 \sum_{i=1}^{N} |\eta_{obs}(f_i) - \eta_0(f_i)| + w_2 \sum_{j=1}^{M} |\eta_{obs}(f_j) - \eta_0(f_j)|, \quad (2)$$

where $\eta_{obs}(f)$ and $\eta_0(f)$ are the observed and predicted RAs, respectively. The first term corresponds to the misfit at the $i$th frequency in the n-RA ($f_1 = fr_1$ and $f_N = fr_2$), whereas the second term is for the $j$-th frequency in the e-RA ($f_1 = 0.04$ Hz and $f_M = fr_1$). We assigned different weights, $w_1 = 0.6$ and $w_2 = 0.4$, because the stacking error of the n-RA was small compared with that of the e-RA. The predicted RA was calculated using DISPER80[45]. First, we estimated the predicted RA with initial Vs profiles. For each station in DONET1, the initial Vp profile was created by referring to a $P$-wave tomographic velocity model along the line, and initial Vs and density profiles were also created using the empirical relationships[22]. For stations in DONET2, we estimated an averaged Vp profile under the seafloor along the line and used it as an initial Vp profile beneath the seafloor at each station.

We defined $v_k$ as the one-dimensional Vs profile at the $k$-th layer, and $\Delta E$ as the energy change. At each iteration step, $v_k$ with a depth interval of 0.1 km is perturbed and the methodology of the velocity structure update follows previous studies on simulated annealing[47–49]. The updated $v'_k$ is given by

$$v'_k = \begin{cases} v_k - \Delta v & \text{if} \quad \alpha < 0.5 \\ v_k + \Delta v & \text{if} \quad \alpha > 0.5 \end{cases}, \quad (3)$$

where $\alpha$ is a random number between 0 and 1. The velocity perturbation is $\Delta v = 0.02$ km s$^{-1}$. To achieve stable DISPER80[45] calculations and continuity in the velocity model, the update was not performed for the layers and the Vs value in the previous model was preserved, for the following two cases: (1) Vs for some layers in the updated $v'_k$ has a value < 0.1 km s$^{-1}$; (2) the velocity difference between Vs in a layer and Vs either above or below the layer, i.e., $\delta v = v_k - v_{k+1}$ or $\delta v = v_k - v_{k-1}$, exceeds 0.2 km s$^{-1}$ from the seafloor to 1 km depth beneath the seafloor and 0.1 km s$^{-1}$ from there to 20 km depth beneath the sea surface, which is equivalent to the maximum velocity gradient of 1 and 2 km s$^{-1}$ per km in depth. In particular, the large criterion of 0.2 km s$^{-1}$ allows us to accept higher velocity gradients near the seafloor. The updated $v'_k$ is accepted when $\Delta E \leq 0$, whereas if $\Delta E > 0$, the acceptance depends on the probability,

$$P = \exp(-\Delta E / T), \quad (4)$$

where $T$ is the temperature. The annealing schedule at the $n$-th step is described as $T_n = \gamma^n T_0$, and we assigned $\gamma = 0.996$ and $T_0 = 3E_0$, where $E_0$ is the result of the first step in Equation (2). If $\alpha \leq P$, where $\alpha$ is also a random number between 0 and 1, the updated $v'_k$ is accepted. When the Vs profile is updated, Vp and density profiles are also updated using the empirical relationships[22] from the Vs profile. As $E$ approximately converges at 2,000 iterations, the iteration stops at 3,000 steps (Supplementary Fig. 7). Supplementary Fig. 8 shows examples of original and optimal velocity models and the corresponding RAs.

In order to estimate the error of the Vs profile in the inversion technique, we prepared 50 RAs at each station with a boot-strapping technique. Selecting a suite of e-RAs from the examined earthquakes through repetition, an e-RA is constructed by selecting median values at all frequencies. Here, the selection of a suite of RAs is performed by generating random number sequences. Similarly, we also prepared 50 n-RAs by selecting a suite of n-RAs from all the observation days and choose median values. In total, we had 50 e-RAs and 50 en-RAs at each station, and connected them at $fr_1$ to prepare 50 en-RAs. Repeating these 3,000 iterations for 50 en-RAs with different random number sequences, we evaluated the standard error of the obtained Vs profiles.

**Synthetic test**. We tested whether the methodology introduced in the previous section is capable of estimating correct velocity models. The initial velocity model is the same as that used in the test for RA sensitivity, i.e., the Vs profile at KMB06. Supplementary Table 3 summarizes the six examined models, G–L, in which different Vs perturbations are assigned to the initial velocity model beneath the seafloor. To test the aforementioned inversion technique, we calculated RAs by changing the $v'_k$ from the initial velocity model and found the velocity model for which the misfit function is a minimum. We confirmed that it properly reproduces the Vs perturbations given in models G–L.

As the perturbations in the models G and H are assigned to shallow depths, the obtained velocity models (red lines in left panels in Supplementary Fig. 8a, b)

appear consistent with the modified (black lines in the panels) and original (dashed lines in the panels) velocity models. However, in addition to these two models, the obtained velocity profiles at depths > 10 km deviate slightly from the modified velocity models in models I and L (Supplementary Fig. 8c, f). These deviations reflect less sensitivity of the RA to deeper structure. In the case of a large perturbation in model J, the modified velocity model could be reproduced (Supplementary Fig. 8d). Interestingly, in the result of model J, a low-velocity zone is also successfully reproduced by the inversion technique used in this study. However, if a velocity reduction is significantly large (model K), such a feature may be underestimated by the inversion technique (Supplementary Fig. 8e). In addition, in cases where the perturbations were assigned at a narrow depth interval (1 km), as in model L, our inversion technique could not reproduce the original perturbed velocity model (Supplementary Fig. 8f). This is because the deep and sharp velocity changes (models L and M) result in small RA changes (Supplementary Fig. 8f, g).

**Estimates of pore pressure ratio**. We employ a method of a previous study[11] that uses a Vp structure for estimating $\lambda^*$, because we can compare the obtained $\lambda^*$ and the fluid pressure with those of the previous study[11]. This approach enables us to perform a calibration of our results for the southeast of the Kii Peninsula and further estimate the relative $\lambda^*$ in the southwest of the Kii Peninsula. As we estimate $\lambda^*$ from Vp profiles, the obtained $\lambda^*$ may be slightly underestimated with respect to that from Vs structure, as indicated by a previous study[21].

In the inversion, we calculated Vp and density profiles from the Vs profiles using the empirical relationships[22]. The Vp profile at the final iteration step was used to estimate the pore pressure ratio. We used the Vp–porosity relationship[50], and the relationship between porosity and effective mean stress[11]. In Supplementary Fig. 4, we show profiles of pore pressure ratios, with two interpretations of the décollement location along L4[17,18], for a case where the accretionary prism is under the margin-normal horizontal compression, i.e., a stress state in which the margin-normal horizontal compression is the largest, followed in order by margin-parallel horizontal compression and vertical compression. As the inversion technique used in this study cannot reproduce a sharp low-velocity layer, the obtained Vs profile may produce a smoothed low-velocity layer with respect to the real structure, as shown in the synthetic test (Supplementary Fig. 8e). In this case, the maximum $\lambda^*$ value associated with a remarkable velocity reduction is underestimated by 0.1.

**Data availability**. Codes and derived data, including one-dimensional Vs profiles, that support the findings of this study are available upon request. Plots and maps were created by using TheGeneric Mapping Tools (GMT)[51].

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

## Acknowledgements

We thank discussion and constructive comments from S. Kodaira, Y. Fukao, and O. Kuwano. We are grateful for Y. Asano and H. Sugioka for providing us the sVLFE catalogs, and A. Nakanishi and K. Takizawa for providing us the Vp velocity model. We appreciate people relevant to the development and maintenance for DONET. Comments from Gerhard Pratt greatly improved the manuscript. This work is supported by a Research Fellowship of the Japan Society for the Promotion of Science (JSPS) for Grants-in-Aid for Young Scientists (B) (15K17753). Some plots, including maps, were made by using The Generic Mapping Tools (GMT).

## Author contributions

T.T. designed the study, performed calculations, and wrote the paper with contributions from all co-authors. E.A. and T.K. organized the data acquisitions and supported the data processing on pressure records. T.N. and K.S. organized the instrumental and other data necessary for the processing. M.N. adapted sVLFE data to the velocity structure.

## Additional information

**Competing interests:** The authors declare no competing financial interests.

