## [Peer Review File · Nature Communications]

Reviewers' comments:

Reviewer #1 (Remarks to the Author):

In my opinion this manuscript represents a significant and important scientific advance, and it is suitable for publication in Nature Communications following appropriate revision.

I am attaching a separate report (PDF) with my detailed comments for the authors consideration when preparing their report.

R. Gerhard Pratt
University of Western Ontario

Reviewer #2 (Remarks to the Author):

This study investigates seismic velocity structures within the Nankai accretionary prism toe using a relatively novel method, an inversion of Rayleigh admittance, with data from a uncommonly-dense offshore network. The quality of the analysis is almost satisfactory: results have been double-checked by using different data set (i.e., earthquake records vs. the combination of earthquake and ambient noise records), and have been justified by sensitivity tests. The finding, spatial correlation between low-velocity zones (LVZs) and shallow very low-frequency earthquakes (sVLFE) across a wide range (> 200 km) of a subduction zone, is also new and important to better understand generation mechanisms of sVLFEs. Since the offshore seismic monitoring including both displacement and pressure observations is a developing hot topic in the field of seismology, the authors' new attempt - an exploration of LVZs by Rayleigh admittance - will draw much attention of readers.

Based on the reason above, I think this paper worthy of publication. There is, however, an issue to be addressed before publication regarding pore pressure ratio estimation. Also, some presentations of figures and text should be improved to make them easy to understand.

Major comments:

On possible bias of velocity estimation

In Figure 2, the authors present LVZ with ~5 km thickness (e.g., the most landward station along L1). The thickness seems exaggerated, compared with that of the previous studies (~2 km at most, Lines 50-52). In my opinion, this reflects that the LVZ is thick enough to be detected by Rayleigh admittance, but that the velocity contrast is too sharp to be recovered as the original thickness under the smoothing constraint. In such a case, the resultant velocity should be biased such that the magnitudes of velocity anomalies are reduced. My concern is that this bias leads to the underestimation of pore pressure ratio. Is it possible to evaluate this possible artificial bias on Vs through synthetic tests? I think rough estimation is acceptable for the authors' conclusion. Otherwise, the authors may choose not to estimate pore pressure ratio. I think this does not lower the value of this paper because the spatial correlation between sVLFE activity and LVZ itself is a new finding.

Minor comments

Lines: 150-155:

These sentences are bit confusing. In my understanding, the authors divide regions into three groups by λ^* (let's say group A-C): group A with $\lambda^*=0.35-0.49$ (low), group B with $\lambda^*=0.54-0.59$ (intermediate), and group C with $\lambda^*=0.63-0.73$ (high). The authors refer to these groups A-C as "weak (Lines 151)", "very weak (Line 152)", and "most very weak (Lines 151-152)" volumes, respectively, and further state that "very weak" volume is linked to the low sVLFE activities (Line

154-155). Since sVLFE has been believed to occur on weak faults, the last sentence associating low sVLFE activity with "very weak" volume is confusing. I recommend the authors to label these groups in a more straightforward way, like "low", "intermediate", and "high", for example. I am also a little doubtful on the statement in Line 150-151. Is there any reason to refer $\lambda^*=0.35-0.49$ as "weak" volume?

Line 156:

"carries" should be "carry".

Line 199:

"appear" should be "appears".

Line 200:

"distanct" should be "distinct".

Line 373:

What is the definition of E_0 ?

Fig. 1:

Is it possible to add node names (KMA, KMB, etc.) to this figure? Since the node names are cited many times in the main text, it is more convenient if station nodes appear on this figure.

Fig. 3:

What are the focal depths of sVLFEs shown by yellow circles? Are there any clues indicating that the sVLFEs occur 5-10 km depth even far from trench? If not, the comparison of dVs with these hypocenters may have no meaning for landward stations (MRA, MRB, MRC, KME, and KMA). In my guess, the yellow hypocenters are important only for demonstrating low sVLFE activity beneath MRD and MRE. Current presentation of Figure 3 will lead to misunderstanding of readers.

Fig. 4:

Is it possible to show hypocenters of sVLFE in profiles L1-L3?

Fig. 5:

Why are sVLFEs (i.e., beach balls) not shown in the path-like LVZs where the authors assume high sVLFE activity but shown in the area of low sVLFE activity?

Caption of Fig. S5 (c):

Should "(c) Obtained and initial Vs profiles" be "(c) Obtained (black) and initial (red) Vs profiles"?

Caption of Fig. S6:

"mode (G)" should be "model (G)"

Reviewer #3 (Remarks to the Author):

This paper uses the new DONET cabled ocean bottom instrument array to estimate seismic velocity structure along the frontal part of the Nankai subduction zone. It reveals a widespread low velocity zone (LVZ) along what they interpret to be the megathrust. What is novel here is the use of the new, fantastic DONET array of seismometers and pressure gauges to determine shear wave velocity (Vs) structure over a broad area, using the Rayleigh Admittance technique. This is useful and interesting, and clearly the seismology is the strong part of the paper. It is accompanied by a series of useful and necessary sensitivity and resolution tests. From Vs they estimate relative pore pressure within the thrust zone, following previously published calibrations. That part is weaker,

and could use some work. Comments follow.

1. Although the seismic observations are novel the pore pressure structure inferred is not a big surprise and not as novel as it could be, as it generally repeats previous work. Ito and Obara proposed such an association between low-velocity layers and VLFE locations. Park et al. 2010 showed the LVZ in Vp images under the DONET1/Kumano Basin area, and Kitajima and Saffer 2012 relate it to high pore pressures and thus to VLFE mechanisms through laboratory calibration of Vp-porosity-pressure data. In fact the current authors just use the Kitajima and Saffer calibration to estimate pressure from Vp. The main novelty here is the use of admittance between Rayleigh waves on the ground and in pressure records to estimate Vs instead of Vp. To some extent Vs should be more sensitive to the poroelastic effects that relate velocities to porosity and pore pressure, so this should be a better way. It would be useful to see why Vs is providing something different than Vp, perhaps through some direct comparisons where both data types exist. Such a comparison could help test whether or not the models of pressure-Vp are appropriate.

More specifically, the basic assumption in calculating pore pressure is that the calibrations and logic of Kitajima and Saffer apply. The logic: (a) a unique relationship between Vp and porosity, laboratory calibrated from drill cores; (b) the porosity is completely controlled by the effective mean stress (mean stress – pore pressure) also in a manner calibrated in the lab; (c) the stress state or relationship between vertical and mean stress is known; and (d) assuming a density for the solid material one can estimate effective stress expected for hydrostatic conditions. While there is growing evidence that an approach like this makes sense in the shallow subsurface, it is a purely poroelastic, soil-mechanics approach that could break down easily. For instance it assumes that lithological variations are unimportant (perhaps OK given that the calibration samples also come from the Nankai margin) and that compaction is reversible – i.e. cementation and alteration do not occur. Metamorphism introduces further irreversible changes to pore structure. While I could see this being true in the shallow wedge it seems less likely farther from the trench, or where oceanic basement is involved. This assumption needs to be tested somehow. One useful test would be to compare with Vs seaward of the trench, to make sure that undeformed sediments have velocities expected for hydrostatic conditions. The extra step of converting Vs to Vp (line 415-416) adds additional uncertainty, since those conversions are nonunique. It could be tested by a direct quantitative comparison with the Park et al. (2010) Vp image, which is in the DONET1 volume.

Overall, accepting the Kitajima and Saffer calibrations at face value, and not using the novel Vs data to explore the problem further, significantly reduces the novelty and rigor of the paper.

2. I don't see how "belt-like" and "patch-like" are different (e.g., p. 10). The main difference is that station distributions are in small patches in the "patch-like" area (DONET2) but are more widespread in the other. Without null results for stations between the patches one cannot say that pore pressure anomalies are patchy. It looks like station distribution gives an apparent correlation to VLFEs dictated by sampling. Also, some of the "patch-like" areas just south of Kii seem to have only weak LVZ's. Overall I think this discussion of correlation with specific VLFE clusters is fairly weak, and should be removed or rethought.

3. Throughout the paper, the codes for specific OBSs are used to describe data (KMD13, MRG, etc.). However these do not appear on any maps in the main text, making it very difficult to tell what the authors were writing about. This made the paper hard to review, and absolutely must be fixed. Even in the supplement there seem to be 3-letter codes for regions (KMD, MRD, etc) and numbers nearby, but not codes like "KMD13". Fig. S1 has no scale nor coordinates, and it is hard to associate the numbers with the stations. The text should be rewritten to use station names much less, and any that are used should show up on a map in the main text, using the same designations.

4. Although there is clearly a LVZ present, the resulting lambda-* values (pressure ratios) are not

very high. E.g., on p. 8, values of 0.35-0.49 are described as “essentially mechanically weak”. What does “weak” mean here? this term is not well explained, but does not seem consistent with the much higher near-lithostatic pressures inferred in many other settings (values >0.8 say). Overall it looks like the fault here is not very overpressured, although this may be a resolution issue. Again, a direct comparison with Kitajima and Saffer, who analyze the Vp LVZ beneath DONET1, and interpreted λ as high as 0.77. Such a comparison might help resolve whether or not the assumptions are correct and that the model is good for predicting pressure.

5. lines 156-163: this paragraph asserts that because the LVZ is observed >20 km away from the deformation front the fluids must be from dehydration reactions not compaction. Yet, the only way the pressure-porosity model makes sense is if pore geometry is in the compaction regime; if this is true then the pore pressure arguments don't work (see point 1 above). Further, the 20 km cutoff is arbitrary. I would delete this paragraph.

6. The Methods section seems rather long, and needs the Supplemental figures to make sense of it. I would put this all into the Supplement for ease of reading, or just write the whole thing as a longer paper for a long-format journal.

Response to Reviewer #1, Dr. Gerhard Pratt

We appreciate careful reading and comments from the Reviewer #1. We have revised the manuscript with the guidance and constructive comments. The label “L” indicates the line number in the revised manuscript.

Line 1: I am not sure the title adequately conveys the meaning the authors intend. Perhaps they want to consider using the term “shallow very low frequency earthquakes” in the title?

Response:

Thanks for your suggestion. We changed the title, as “Sporadic low-velocity volumes spatially correlate with shallow very low frequency earthquake clusters.”

Lines 18-32 (Abstract): This is not a particularly well-written abstract. I would suggest a number of items be considered:

- The abstract does not make it clear that Rayleigh wave displacement-pressure ratios are used to extract S-wave velocity profiles using individual sea-floor stations
- The abstract does not make it clear that Simulated Annealing is used to invert the RA measurements for S-wave profiles
- The abstract claims the distribution of LVZs “possibly” correlates with the sVLFs. This is too ambiguous, and the authors need to be more explicit and more confident about the nature of the correlation they observe.

Response:

Thanks for your comments on abstract. We added descriptions on what we did and used, and we explicitly state what we found in the revised manuscript (L26).

Line 27: Change the word “obtained” to “estimated”.

Response: Thanks. We changed it.

Line 49: Pluralise the word LVZ (to read LVZs), since you are talking about a world wide distribution of these zones.

Response: Thanks. We changed it.

Line 53: Awkward sentence construction. I suggest something like this: “The V_p values vary from ... within the LVZ and the overlying layers ...”

Response: We edited it, as you suggested (L53).

Line 56: I checked reference [11] (Moore et al., 2009) but I could find no indication of the claim that the “Nankai region is thought to be a mechanically weak volume at depth”.

Response: Many thanks for pointing out the wrong citation. We corrected it (L56).

Line 63: The claim that the LVZ is “only resolve[d]” along the margin-normal direction should be reconsidered. In fact, Park et al. (2010) discuss lateral variations (along the margin-parallel direction) through the use of a range of 2-D profiles distributed along an approximate range of locations covering 100 km parallel to the margin.

Response: Park et al. (2010) estimated a 3D V_p model for 12 km x 62 km in horizontal space, and compared it with a 2-D reflection profile (not velocity model). As a result, the low-velocity region (3D velocity model) corresponds to a transparent region in the 2D reflectivity profile. Based on the fact, Park et al. (2010) speculate the locations of the LVZ from many 2D reflectivity profiles (not from velocity model, but only from reflectivity). This means that the LVZ is now resolved by a 3D V_p model with 12 km wide along trough-parallel direction (Park et al. 2010) and a 2D V_p model (Kamei et al. 2012).

Line 76: The authors introduce the term “Rayleigh Admittance” and provide citation [15], which is to Kaneda et al. (2015) (who do not discuss RA at all). In fact the authors owe their use of Rayleigh waves displacement vs pressure ratios to Ruan et al. (2014), and they should insert a citation at this stage.

Response: Many thanks for pointing out the wrong citation. We corrected them (L76).

Line 82 and 88: The authors are using the past tense at Line 82, and the present tense at Line 88. Obviously they should avoid mixing tenses - my personal preference would be that they use the past tense to describe their analysis, and the present tense for their discussion and conclusions.

Response: Thanks. We corrected it.

Line 95: The authors do not need to say “see Methods for the definition of the LVZ feature”, because in fact their definition is stated in the very next sentence.

Response: We removed the terms in the revised manuscript.

Line 99: I did not follow how the definition of the LVZ could include condition (2), that VS should be constant as a function of depth. Clearly some re-writing is required to clarify.

Response: What we would like to mention is that, if a V_s profile shows a velocity reduction greater than 20% in the depth range of 5–10 kmbsl from a reference velocity model, we consider it as the LVZ feature, irrespective of velocity gradient. We changed the description (L95).

Line 110: The location of the station and node names is so central to the reading of the paper that these should not be relegated to supplementary Figure 1, but should somehow be incorporated into the main Figure 1.

Response: Many thanks for your good suggestion. We added some labels for node names, which are often stated in the main text.

Line 134: The authors refer in citation [13] to “the method of a previous study”, but this does not make it clear that they are actually using empirical relations here to estimate V_p from their V_s profiles. This could be stated, and they should insert their citation [44] here as well.

Response: Thanks for your good suggestion. In the revised manuscript, we explicitly state that we used a method of Kitajima and Saffer (2012) with an empirical relation (Brocher 2005) (L143–144).

Line 151: Change the word “obtained” to “recovered”.

Response: Thanks. We changed it.

Line 154: The authors are stating a discussion point here regarding the spatial correlation of the LVZs with the sVLFs. This point needs more careful consideration and I recommend they simply omit this sentence here, and save the discussion for the

section to come.

Response: Thanks. We removed the sentence.

Line 169: Clarify the mechanism by which fluids “could be trapped there”. Are you speculating that the shear fractures could trap fluids? If yes, say so, and if not be more explicit.

Response: We changed the description.

Line 175: Rather than stating unequivocally that the activity is distributed “sporadically”, the authors should source this claim by crediting the literature they claim, i.e., by writing

“The activity has been observed to be spatially distributed in a sporadic manner, with some clusters [3,24]. More importantly, the authors seem to be hedging as to whether their term “sporadic” should be interpreted to mean spatially or temporally. It seems critical that the authors are as clear as possible here.

Response: Thank you very much for your good suggestion. As you suggested, we changed the description (L179–180).

Line 196: The authors introduce the terms “patch-like” and “belt-like”. It was not immediately clear to me what these terms meant, and what criteria were used to discern whether a given distribution belong to which type of volume. These clearly subjective terms require some kind of description and/or definition.

Response: We agree with your comment. We removed the description on patch and belt, and preserved only patch in the revised manuscript.

Line 210: The authors write about their own paper: “... this study is the first to reveal that ...LVZs ... are correlated ..., and interprets that sVLFs may be activated ...”. This is the kernel of the authors major claim to scientific significance, and the sentence is impossibly passive and inconclusive. The authors seem to be lacking the confidence to write in the active voice here and to demonstrate any confidence in their own conclusion. Certainly they should not be stating that their own paper “interprets” this conclusion, rather they should state “... we interpret therefore that sVLFs may be activated ...”.

Response: We used “we reveal ... and interpret” in the revised manuscript (L225–227).

Line 231: This is the beginning of the methodology section, and the authors are referring to the Rayleigh admittance analysis they use, which is clearly based on the work of Ruan et al. (2014). There should clearly be a citation at this line.

Response: We agree with your comment. We cited it at appropriate places in the revised manuscript.

Line 242: If possible, include the instrument manufacturer, the instrument model and a citation to the specifications or manual (if at all possible).

Response: Thanks for the comment on instrument response. In the Method section, we added descriptions on seismometer and APG used in this study. On DPG, we only have a paper in Japanese. In this paper, Abstract and figure captions are written in English, so if you are interested, please visit to the following URL.

https://www.jstage.jst.go.jp/article/jamstecr/2009/0/2009_0_141/_article/cited-by
(L255–258)

Line 264: Equation (1) clearly comes from Ruan et al. (2014), and should be cited appropriately.

Response: We cited it near equation (1) in the revised manuscript.

Line 327: Change “becomes low” to “is reduced”.

Response: We changed it. Thanks.

Line 446: Bibliography item [6] appears to be missing a title.

Response: Thanks. We corrected it.

Line 548: There appears to be a typo in the title.

Response: We removed “L”, and replaced to “.”. Thanks.

References

- Brocher, T. M. Empirical relations between elastic wavespeeds and density in the Earth's crust. *Bull. Seis. Soc. Am.* **95**, 2081–2092 (2005).
- Kamei, R., Pratt, R. G. & Tsuji, T. Waveform tomography imaging of a megasplay fault system in the seismogenic Nankai subduction zone. *Earth. Planet. Sci. Lett.* **317–318**, 343–353 (2012).
- Kitajima, H. & Saffer, D. M. Elevated pore pressure and anomalously low stress in regions of low frequency earthquakes along the Nankai Trough subduction megathrust. *Geophys. Res. Lett.* **39**, L23301 (2012).
- Park, J. O., Fujie, G., Wijerathne, L., Hori, T. & Kodaira, S. et al. A low-velocity zone with weak reflectivity along the Nankai subduction zone. *Geology*, **38**, 283–286 (2010).

Response to Reviewer #2

We appreciate the comments and suggestions from the Reviewer #2, particularly for the comment on bias and synthetic tests. We have revised the manuscript with the guidance and constructive comments. The label “L” indicates the line number in the revised manuscript, except for L1–L4 in Fig. 1.

Major comments:

On possible bias of velocity estimation

In Figure 2, the authors present LVZ with ~5 km thickness (e.g., the most landward station along L1). The thickness seems exaggerated, compared with that of the previous studies (~2 km at most, Lines 50-52). In my opinion, this reflects that the LVZ is thick enough to be detected by Rayleigh admittance, but that the velocity contrast is too sharp to be recovered as the original thickness under the smoothing constraint. In such a case, the resultant velocity should be biased such that the magnitudes of velocity anomalies are reduced. My concern is that this bias leads to the underestimation of pore pressure ratio. Is it possible to evaluate this possible artificial bias on Vs through synthetic tests? I think rough estimation is acceptable for the authors' conclusion. Otherwise, the authors may choose not to estimate pore pressure ratio. I think this does not lower the value of this paper because the spatial correlation between sVLF activity and LVZ itself is a new finding.

Response: Many thanks for your suggestion on the bias. We added one more synthetic test that attempts to estimate a sharp and extreme (40%) low-velocity layer (Supplementary Fig. 8e). As a result, we obtained a Vs profile smoothing the velocity reduction (underestimated), and also obtained a lower λ^* value compared with the original one. This result is predicted by your comment. We added explanations (L424–426, L448–452).

Minor comments

Lines: 150-155:

These sentences are bit confusing. In my understanding, the authors divide regions into three groups by λ^* (let's say group A-C): group A with $\lambda^*=0.35-0.49$ (low), group B with $\lambda^*=0.54-0.59$ (intermediate), and group C with $\lambda^*=0.63-0.73$ (high). The authors

refer to these groups A-C as “weak (Lines 151)”, “very weak (Line 152)”, and “most very weak (Lines 151-152)” volumes, respectively, and further state that “very weak” volume is linked to the low sVLFE activities (Line 154-155). Since sVLFE has been believed to occur on weak faults, the last sentence associating low sVLFE activity with “very weak” volume is confusing. I recommend the authors to label these groups in a more straightforward way, like “low”, “intermediate”, and “high”, for example. I am also a little doubtful on the statement in Line 150-151. Is there any reason to refer $\lambda^*=0.35-0.49$ as “weak” volume?

Response: We agree with your comment. In the revised manuscript, we removed the description on “weak” and the others, and only mentioned the variation of λ^* .

Line 156:

“carries” should be “carry”.

Response: Thanks. We changed it.

Line 199:

“appear” should be “appears”.

Response: Thanks .We changed it.

Line 200:

“distanct” should be “distinct”.

Response: Thanks .We edited it.

Line 373:

What is the definition of E_0 ?

Response: Thank you. E_0 is the result at the first step in eq. (2), and we added explanation in the revised manuscript.

Fig. 1:

Is it possible to add node names (KMA, KMB, etc.) to this figure? Since the node names are cited many times in the main text, it is more convenient if station nodes appear on this figure.

Response: We added a part of them in Figs. 1 and 3, which is often referred in the main

text (L388).

Fig. 3:

What are the focal depths of sVLFs shown by yellow circles? Are there any clues indicating that the sVLFs occur 5-10 km depth even far from trench? If not, the comparison of dVs with these hypocenters may have no meaning for landward stations (MRA, MRB, MRC, KME, and KMA). In my guess, the yellow hypocenters are important only for demonstrating low sVLF activity beneath MRD and MRE. Current presentation of Figure 3 will lead to misunderstanding of readers.

Response: Thanks for the comment on the focal depth. Sugioka et al. (2012) determined the focal depths of sVLFs using a realistic velocity model and seafloor records, and obtained focal depths of 5–11 km within the accretionary prism toe. We referred this result, and speculate that sVLFs determined from land-based observations occur in the same manner. We added explanation in the revised manuscript (L192).

Fig. 4:

Is it possible to show hypocenters of sVLF in profiles L1-L3?

Response: At present, we cannot. We only have reliable focal depths of sVLFs in Sugioka et al. (2012), because it uses a realistic velocity model for the hypocentre determination.

Fig. 5:

Why are sVLFs (i.e., beach balls) not shown in the path-like LVZs where the authors assume high sVLF activity but shown in the area of low sVLF activity?

Response: We added beach balls in the other patches (Fig. 4 in the revised manuscript).

Caption of Fig. S5 (c):

Should “(c) Obtained and initial Vs profiles” be “(c) Obtained (black) and initial (red) Vs profiles”?

Response: Thanks. We corrected it.

Caption of Fig. S6:

“mode (G)” should be “model (G)”

Response: Thanks. We corrected it.

Reference

Sugioka, H., Okamoto, T., Nakamura, T., Ishihara, Y. & Ito, A. et al. Tsunamigenic potential of the shallow subduction plate boundary inferred from slow seismic slip. *Nature Geoscience* **5**, 414–418 (2012).

Response to Reviewer #3

We appreciate the comments and suggestions from the Reviewer #3, particularly for the comment on the use of V_s for estimating the pore pressure. We have revised the manuscript with the guidance and constructive comments. The label “L” indicates the line number in the revised manuscript, except for L1–L4 in Fig. 1.

1. Although the seismic observations are novel the pore pressure structure inferred is not a big surprise and not as novel as it could be, as it generally repeats previous work. Ito and Obara proposed such an association between low-velocity layers and VLFE locations. Park et al. 2010 showed the LVZ in V_p images under the DONET1/Kumano Basin area, and Kitajima and Saffer 2012 relate it to high pore pressures and thus to VLFE mechanisms through laboratory calibration of V_p -porosity-pressure data. In fact the current authors just use the Kitajima and Saffer calibration to estimate pressure from V_p . The main novelty here is the use of admittance between Rayleigh waves on the ground and in pressure records to estimate V_s instead of V_p . To some extent V_s should be more sensitive to the poroelastic effects that relate velocities to porosity and pore pressure, so this should be a better way. It would be useful to see why V_s is providing something different than V_p , perhaps through some direct comparisons where both data types exist. Such a comparison could help test whether or not the models of pressure- V_p are appropriate.

More specifically, the basic assumption in calculating pore pressure is that the calibrations and logic of Kitajima and Saffer apply. The logic: (a) a unique relationship between V_p and porosity, laboratory calibrated from drill cores; (b) the porosity is completely controlled by the effective mean stress (mean stress – pore pressure) also in a manner calibrated in the lab; (c) the stress state or relationship between vertical and mean stress is known; and (d) assuming a density for the solid material one can estimate effective stress expected for hydrostatic conditions. While there is growing evidence that an approach like this makes sense in the shallow subsurface, it is a purely poroelastic, soil-mechanics approach that could break down easily. For instance it assumes that lithological variations are unimportant (perhaps OK given that the calibration samples also come from the Nankai margin) and that compaction is reversible – i.e. cementation

and alteration do not occur.

Metamorphism introduces further irreversible changes to pore structure. While I could see this being true in the shallow wedge it seems less likely farther from the trench, or where oceanic basement is involved. This assumption needs to be tested somehow. One useful test would be to compare with V_s seaward of the trench, to make sure that undeformed sediments have velocities expected for hydrostatic conditions. The extra step of converting V_s to V_p (line 415-416) adds additional uncertainty, since those conversions are nonunique. It could be tested by a direct quantitative comparison with the Park et al. (2010) V_p image, which is in the DONET1 volume.

Overall, accepting the Kitajima and Saffer calibrations at face value, and not using the novel V_s data to explore the problem further, significantly reduces the novelty and rigor of the paper.

Response: Thank you very much for your comment on the pore pressure, and we agree with the importance of V_s for accurately estimating the pore pressure ratio (λ^*) (or poroelastic property). At present, Tsuji et al. (2014) only estimates the distributions of λ^* in the Nankai accretionary prism from two velocity structures (V_p and V_s). According to the paper, although the estimated λ^* from V_s is slightly higher than that from V_p , they are similar to each other, and they are consistent with that from Kitajima and Saffer (2012).

The reason we avoid to estimate λ^* with V_s structure are based on the following point. Tsuji et al (2011) found an anisotropic structure in shear wave speed within the Nankai accretionary prism, possibly indicating the anisotropy within the LVZs. Because the V_s variation as a function of polarization direction affects the estimate of λ^* , it is necessary to reveal the degree of anisotropy on the LVZs with other seismic approaches, e.g., receiver function. Moreover, according to Tsuji et al. (2014), the use of V_s requires an assumption, i.e., effective-pressure coefficient (equations 2 in Tsuji et al. 2014). More investigations on the assumption would be necessary for assessing the accuracy and error of the obtained λ^* . We would like to wait for (or partly estimate) such additional information, and take into account them when we use our V_s results for estimating λ^* in future study.

In our paper, we would like to primarily argue the spatial relationship between the LVZs and sVLFE clusters. On the λ^* , we calibrate it at a location (southeast of the

Kii Peninsula) by using a method of Kitajima and Saffer (2012), and apply the method to other locations (southwest of the Kii Peninsula) for obtaining relative λ^* values. As a result of the calibration, we could obtain a consistent result with that of Kitajima and Saffer (2012). In the revised manuscript, we modified the following points.

- 1) We noted the previous studies on the λ^* estimation from V_p and V_s , by referring to Kitajima and Saffer (2012) and Tsuji et al. (2014) (see descriptions in L136, L433).
- 2) We also attempted to estimate the λ^* using our results and the method from Kitajima and Saffer (2012), and compared the result to that from the previous study for a calibration at L4. Then we estimated the (relative) λ^* at the other lines (L1–L3) (see L143–160).
- 3) We added a description on a slight underestimation of our results due to the use of V_p for estimating λ^* (L437).
- 4) We moved the figure on λ^* profiles from the main text to Supplementary Information.

Moreover, we thank you to raise a main novelty of this study, i.e., the use of the displacement-pressure ratio of the Rayleigh wave propagation. Because such descriptions may be insufficient in the previous version of the manuscript, we added those in Abstract and Discussion sections.

2. I don't see how "belt-like" and "patch-like" are different (e.g., p. 10). The main difference is that station distributions are in small patches in the "patch-like" area (DONET2) but are more widespread in the other. Without null results for stations between the patches one cannot say that pore pressure anomalies are patchy. It looks like station distribution gives an apparent correlation to VLFs dictated by sampling. Also, some of the "patch-like" areas just south of Kii seem to have only weak LVZ's. Overall I think this discussion of correlation with specific VLF clusters is fairly weak, and should be removed or rethought.

Response: In the revised manuscript, we removed the descriptions of patch and belt, and added some explanations on patch-like LVZs. Also, we added descriptions on patch-like LVZ (L198–203).

3. Throughout the paper, the codes for specific OBSs are used to describe data (KMD13,

MRG, etc.). However these do not appear on any maps in the main text, making it very difficult to tell what the authors were writing about. This made the paper hard to review, and absolutely must be fixed. Even in the supplement there seem to be 3-letter codes for regions (KMD, MRD, etc) and numbers nearby, but not codes like “KMD13”. Fig. S1 has no scale nor coordinates, and it is hard to associate the numbers with the stations. The text should be rewritten to use station names much less, and any that are used should show up on a map in the main text, using the same designations.

Response: Many thanks for your comment on the node name. We added labels of KMB, KMD, MRD, MRE, MRF, MRG in Figs. 1 and 3, which are often referred in the main text.

4. Although there is clearly a LVZ present, the resulting λ^* values (pressure ratios) are not very high. E.g., on p. 8, values of 0.35-0.49 are described as “essentially mechanically weak”. What does “weak” mean here? this term is not well explained, but does not seem consistent with the much higher near-lithostatic pressures inferred in many other settings (values >0.8 say). Overall it looks like the fault here is not very overpressured, although this may be a resolution issue. Again, a direct comparison with Kitajima and Saffer, who analyze the Vp LVZ beneath DONET1, and interpreted λ^* as high as 0.77. Such a comparison might help resolve whether or not the assumptions are correct and that the model is good for predicting pressure.

Response: We agree with your comment. In the revised manuscript, we removed the descriptions on “weak”, and noted the λ^* values from this study and the previous study (Kitajima and Saffer, 2012) for a comparison (L136).

5. lines 156-163: this paragraph asserts that because the LVZ is observed >20 km away from the deformation front the fluids must be from dehydration reactions not compaction. Yet, the only way the pressure-porosity model makes sense is if pore geometry is in the compaction regime; if this is true then the pore pressure arguments don’t work (see point 1 above). Further, the 20 km cutoff is arbitrary. I would delete this paragraph.

Response: Thanks for your comment on the pressure-porosity model. According to Fig. 3b in Saffer and Tobin (2011), the main source of fluid supply changes from horizontal compaction to dehydration reaction at a distance from trench, but both mechanisms are possibly related to fluids in the LVZ. Therefore, we changed description on fluid supply

in the revised manuscript (L161–167).

6. The Methods section seems rather long, and needs the Supplemental figures to make sense of it. I would put this all into the Supplement for ease of reading, or just write the whole thing as a longer paper for a long-format journal.

Response: Thanks for your comment on reading. We would like to keep the current style, because the Method section in the format of Nature Communications looks good for reading (sorry for hard reading for reviewers, and many thanks to you).

References

- Kitajima, H. & Saffer, D. M. Elevated pore pressure and anomalously low stress in regions of low frequency earthquakes along the Nankai Trough subduction megathrust. *Geophys. Res. Lett.* **39**, L23301 (2012).
- Saffer, D. M. & Tobin, H. J. Hydrogeology and mechanics of subduction zone forearcs: Fluid flow and pore pressure. *Annu. Rev. Earth Planet. Sci.* **39**, 157–186 (2011).
- Tsuji, T., Dvorkin, J., Mavko, G., Nakata, N., & Matsuoka, T. et al. Vp/Vs ratio and shear-wave splitting in the Nankai Trough seismogenic zone: Insights into effective stress, pore pressure, and sediment consolidation, *Geophysics*, **76**(3), WA71-WA82 (2011).
- Tsuji, T., Kamei, R. & Pratt, R. G. Pore pressure distribution of a mega-splay fault system in the Nankai Trough subduction zone: Insight into up-dip extent of the seismogenic zone. *Earth Planet. Sci. Lett.* **396**, 165–178 (2014).

REVIEWERS' COMMENTS:

Reviewer #2 (Remarks to the Author):

I am satisfied with the authors' revision, especially for their additional synthetic test with an extremely low-velocity layer. I think the article almost ready to be accepted, but I still have some suggestions regarding hypocenters of sVLFs.

On Section "Spatial relationship between LVZ and sVLF",

#1. What the authors should state first in this paragraph is the spatial correlation between LVZ locations and accurate hypocenters provided by the short-term catalog. Then, the less-accurate long-term catalog should be mentioned only for justifying there is no sVLFs activity near MRD and MRE even for long period. I recommend that the important sentence such as Lines 196-197 should be placed earlier than the description of Fig. S3 (Lines 184-185).

#2. A histogram along the trench-parallel direction may be more appropriate to display the sVLF activity listed in the long-term catalog than epicenters (such as Fig. S3) if the authors think their locations in the trough-normal direction highly unreliable.

Reviewer #3 (Remarks to the Author):

Overall this manuscript is much improved. The redone figures in particular help a great deal in seeing what the authors are discussing. I do not have much to add, most of the reviewer comments have been carefully and adequately addressed.

The paper will need some significant work to fix the English usage. I was unable to get very far in it, and had a bit of a hard time understanding the subtleties. For example I am still confused about how exactly the authors are defining a "LVZ" from lines 95-102; what exactly is the reference, and are they just looking at perturbations to it regardless of whether or not its velocities decrease with increasing depth in an absolute sense? There are many other similar issues; sorry I was unable to flag all of them.

The paper now does a better job of describing the other papers that have tried to estimate overpressure along the Nankai thrust (Tsuji et al.,; Kitajima and Saffer; Park et al.). Unfortunately in doing so it is less obvious what is really new here – the association with VLFs was made by Kitajima and Saffer for instance. The method is new, and there is wider along-strike coverage including at least one region that has few VLFs and no real LVZ so that helps in making the pattern clear. It would be good to strengthen the description of what is unique here.

Response to Reviewer #2

On Section “Spatial relationship between LVZ and sVLFE”,

#1. What the authors should state first in this paragraph is the spatial correlation between LVZ locations and accurate hypocenters provided by the short-term catalog. Then, the less-accurate long-term catalog should be mentioned only for justifying there is no sVLFs activity near MRD and MRE even for long period. I recommend that the important sentence such as Lines 196-197 should be placed earlier than the description of Fig. S3 (Lines 184-185).

Response: Thanks. In this section in the 2nd revised manuscript, we firstly stated on sVLFE epicentres from seafloor observations in Fig. 3 (L186–205), and secondarily noted those from land-based observations in Supplementary Fig. 3 (L206–220).

#2. A histogram along the trench-parallel direction may be more appropriate to display the sVLFE activity listed in the long-term catalog than epicenters (such as Fig. S3) if the authors think their locations in the trough-normal direction highly unreliable.

Response: Thank you very much for an interesting comment. We added a histogram in Supplementary Fig. 3, which will help readers to understand the along-strike variation of sVLFE activities.

Response to Reviewer #3

The paper will need some significant work to fix the English usage. I was unable to get very far in it, and had a bit of a hard time understanding the subtleties. For example I am still confused about how exactly the authors are defining a “LVZ” from lines 95-102; what exactly is the reference, and are they just looking at perturbations to it regardless of whether or not it velocities decrease with increasing depth in an absolute sense? There are many other similar issues; sorry I was unable to flag all of them.

Response: On the LVZ feature, we added explanations in L101–103. Because the reference velocity model reflects a V_s averaged over the entire accretionary prism, the LVZ feature indicates a significantly lower velocity (~20 %) than the averaged V_s . Moreover, we asked an editing company on English usage correction.

The paper now does a better job of describing the other papers that have tried to estimate overpressure along the Nankai thrust (Tsuji et al.; Kitajima and Saffer; Park et al.). Unfortunately in doing so it is less obvious what is really new here – the association with VLFs was made by Kitajima and Saffer for instance. The method is new, and there is wider along-strike coverage including at least one region that has few VLFs and no real LVZ so that helps in making the pattern clear. It would be good to strengthen the description of what is unique here.

Response: Thanks. We added some descriptions in Abstract (L28–29) and also in L254-255, in order to strengthen the uniqueness of this study.